

# Odd family reunion: DNA barcoding reveals unexpected relationship between three hydrozoan species

Lara M. Beckmann, Joan J. Soto-Angel, Aino Hosia and Luis Martell

Department of Natural History, University Museum of Bergen, Bergen, Norway

## ABSTRACT

Knowledge of life histories is crucial for understanding ecological and evolutionary processes, but for many hydrozoan species only incomplete life cycles have been described due to challenges in linking hydromedusae with their polyp stages. Using a combination of DNA barcoding, morphology, and ecological information, we describe for the first time the polyp stage of *Halopsis ocellata* Agassiz, 1865 and re-describe that of *Mitrocomella polydiademata* (Romanes, 1876). Campanulinid hydroids referable to *Lafoeina tenuis* Sars, 1874 and collected in the same biogeographic region as the type locality of this species are shown to be the polyp stage of these two mitrocomid hydromedusae. The nominal species *L. tenuis* thus is a species complex that includes the polyp stage of medusae belonging to at least two genera currently placed in a different family. Consistent morphological and ecological differences were found between the polyps linked to each of these two hydromedusae, but molecular results suggest that yet other species may have morphologically similar hydroids. Polyps morphologically identified to *L. tenuis* are therefore better referred to as *Lafoeina tenuis*-type until further associations are resolved, particularly when occurring outside of the area of distribution of *H. ocellata* and *M. polydiademata*. Molecular identification integrated with traditional taxonomy is confirmed as an effective approach to link inconspicuous stages of marine invertebrates with hitherto unknown life cycles, especially in often-overlooked taxa. Disentangling the relationships between *L. tenuis*, *H. ocellata*, and *M. polydiademata* lays the ground for future research aimed at resolving the taxonomy and systematics of the enigmatic families Mitrocomidae and Campanulinidae.

## INTRODUCTION

Hydrozoans are widespread but often overlooked components of marine environments (*Gili et al., 1998*; *Bouillon et al., 2006*). They occur both in the benthos and the plankton of all oceans, where they act as predators (*Purcell, 1989*; *Nicholas & Frid, 1999*; *Orejas et al., 2001*), prey (*Arai, 2005*; *Ayala et al., 2018*), symbionts (*Fernandez-Leborans, 2013*; *Montano et al., 2015*), habitat-formers (*Di Camillo et al., 2017*; *Gomes-Pereira et al., 2017*; *Puente-Tapia et al., 2021*), and significantly contribute to bentho-pelagic coupling (*Gili*

Corresponding author
Luis Martell, luis.martell@uib.no

*et al., 1998*). As a group, hydrozoans are well-known for their wide array of life cycle strategies, including the classic textbook example of substrate-bound hydroid polyps that produce free-swimming hydromedusae (*Boero et al., 1997*; *Bouillon et al., 2006*). Because benthic hydroids and their corresponding medusae have different ecological niches, complete knowledge of their life cycle is crucial to understand the ecological and evolutionary processes underlying species diversity and diversification (*Boero et al., 1997*).

Documenting the entire life cycle of hydrozoans is a challenging task, complicated by hydroids and hydromedusae having been traditionally studied by different groups of researchers (*Boero, Bouillon & Piraino, 1992*). Hydrozoan species have often been described based on a single life stage, which has led to the development of parallel taxonomies with separate names for polyps and their respective medusae (*Brooks, 1886*; *Cornelius, 1977*). Many of these names are still in use, and the process of unifying these classification systems is far from finished (*Schuchert, Hosia & Leclére, 2017*; *Maggioni et al., 2021*). Even when the relationships between polyps and medusae are known, some stages have never been observed in nature and their morphology is described solely based on animals reared in laboratory and/or for juvenile or immature specimens (*Russell, 1953*; *Bouillon et al., 2006*). Hydrozoans have a high degree of phenotypic plasticity and lab-reared animals may be strikingly different from those collected in the field, depending on the conditions they have been exposed to (*Dudgeon & Buss, 1996*; *Meroz-Fine et al., 2003*; *Griffith & Newberry, 2008*). In the past, rearing experiments were the only way to link different life stages in Hydrozoa (*e.g.*, *Rees & Russell, 1937*; *Edwards, 1973a*, *1973b*; *Widmer, 2004*; *Migotto & Cabral, 2005*); however, culture procedures are time-consuming and not always successful (*e.g.*, obtaining only immature specimens) (*Martin, Chia & Koss, 1983*; *Freeman & Ridgway, 1990*). Molecular species identification methods such as DNA barcoding offer an alternative solution to this problem by providing a rapid approach to correlate separate life stages in siphonophore, anthoathecate and leptothecate hydrozoans (*Schuchert, Hosia & Leclére, 2017*; *Schuchert, 2016*; *Pyataeva et al., 2016*; *Grossmann, Lindsay & Collins, 2013*; *Grossmann, Collins & Lindsay, 2014*). Despite recent advancements, our knowledge of hydrozoan life stages is still one of the least complete in all Cnidaria (*Boero et al., 1997*).

Among the Hydrozoa, Mitrocomidae *Haeckel, 1879* and Campanulinidae *Hincks, 1868* are two of the taxa with long-lasting taxonomic confusion due to poorly-known life cycles (*Cornelius, 1995a*). Mitrocomidae is a medusa-based taxon, *i.e.*, it is defined based on morphological characters of the hydromedusa, a stage absent or unknown for most of the campanulinids (*Bouillon et al., 2006*). Campanulinidae, on the other hand, is defined based on characters present in the polyp stage, which is unknown for the majority of the mitrocomids (*Bouillon et al., 2006*). When known, most mitrocomid polyps are indistinguishable between species and—with the exception of three species in genera *Cyclocanna* and *Earleria*—they are described as "*Cuspidella*-type" (*Cornelius, 1995a*; *Widmer, Cailliet & Geller, 2010*; *Schuchert, Hosia & Leclére, 2017*), a morphological facies referable to Campanulinidae (*Cornelius, 1995a*). These polyps are also impossible to differentiate from similar *Cuspidella*-type polyps belonging to other hydrozoan families such as Laodiceidae and Tiaropsidae (*Cornelius, 1995a*; *Bouillon et al., 2006*).

Campanulinid hydroids thus pose numerous taxonomic problems and inconsistencies, as the family has traditionally been used as a catch-all taxon for hydroids that release medusae referable to other hydrozoan taxa (*Cornelius, 1995a*; *Bouillon et al., 2006*). The complicated relationship between Mitrocomidae and Campanulinidae is only partially understood, as several—but not all—campanulinid hydroids produce mitrocomid hydromedusae (*e.g.*, *Schuchert, Hosia & Leclére, 2017*), and many medusa-based and polyp-based species in these two families are in need of a description of their complete life cycle (*Cornelius, 1995a*).

In this contribution, we employ an integrative approach combining morphological, molecular, and ecological information to uncover the connection between the campanulinid polyps of *Lafoeina tenuis Sars, 1874* and the mitrocomid hydromedusae of *Halopsis ocellata Agassiz, 1865* and *Mitrocomella polydiademata* (*Romanes, 1876*). We redescribe the polyp stage of *M. polydiademata* and present a re-evaluation of the life cycle and taxonomy of these three leptothecate species to resolve part of the incongruences in these hydrozoan taxa.

## MATERIALS AND METHODS

### Sampling and DNA work

Individual hydromedusae (*Mitrocomella polydiademata* and *Halopsis ocellata*) and hydroid colonies (*Lafoeina tenuis*) were collected during several sampling events at multiple locations in Norway as part of the Norwegian Taxonomy Initiative projects "Hydrozoan pelagic diversity in Norway (HYPNO)" and "Norwegian marine benthic Hydrozoa (NorHydro)" (Table 1, Fig. 1). The benthic hydroids were either collected with an Agassiz trawl, beam trawl, RP-sledge, Van Veen grab, triangular dredge, or by hand while scuba diving. The plankton samples were collected with either a modified WP3 (750 or 1,000 µm mesh size, non-filtering cod-end), MOCNESS (180 µm), or MIK plankton net. Live hydromedusae were carefully picked from the samples using a light table immediately after collection, and the morphology of each individual was documented with photographs prior to fixation in 96% EtOH. For each hydromedusa and polyp colony, photographs and associated metadata were assembled into electronic vouchers (e-vouchers) connected to the physical specimens deposited in the Invertebrate Collections of the University Museum of Bergen (UMB).

DNA was extracted from 2–3 mm$^3$ of soft tissue, selecting either a section of the umbrella margin (for hydromedusae) or 3–6 polyps (for benthic colonies). The samples were either further processed at the DNA lab of the University of Bergen (UiB) or sent to the sequencing facilities of the Canadian Centre for DNA Barcoding (CCDB—Centre for Biodiversity Genomics, University of Guelph). All samples sent to CCDB were processed according to the protocols described by *Ratnasingham & Hebert (2007)*. At UiB, DNA was extracted using the QuickExtract™ DNA Extraction Solution Kit following the protocol described by *Nygren et al. (2018)*. The mitochondrial molecular markers COI and 16S were subsequently amplified for each specimen following the specifications in Table 2. All PCR products were checked by electrophoresis on 1% agarose gels and those that yielded

**Table 1 List of specimens included in the analysis.** List of specimens of *Mitrocomella polydiademata* and *Halopsis ocellata* from NorHydro/HYPNO, and additional sequences used in the analysis from BOLD, GenBank and the University Museum Bergen (UMB). When available, the accession number for 16S and COI are listed. The specimen life stage (LS) is indicated as M (Medusa) or P (polyp).

| Catalogue number (ZMBN) | Definitive ID | Initial ID | LS | Location | Lat/Long | Depth (m) | Substrate | Source | 16S | COI |
|---|---|---|---|---|---|---|---|---|---|---|
| 150925 | *Halopsis ocellata* | *Lafoeina tenuis* | P | Haltenbanken | 64.813 N 8.970 E | 190 | Polychaete | This study | OP951085 | OP945752 |
| 150926 | *Halopsis ocellata* | *Lafoeina tenuis* | P | Haltenbanken | 64.430 N 8.826 E | 201 | Porifera | This study | OP951086 | OP945753 |
| 150927 | *Halopsis ocellata* | *Lafoeina tenuis* | P | Haltenbanken | 64.430 N 8.826 E | 201 | Polychaete | This study | OP951087 | OP945754 |
| 150928 | *Halopsis ocellata* | *Lafoeina tenuis* | P | Haltenbanken | 64.589 N 8.559 E | 187 | Polychaete | This study | OP951088 | OP945755 |
| 150929 | *Halopsis ocellata* | *Lafoeina tenuis* | P | Haltenbanken | 64.971 N 8.354 E | 210 | Polychaete | This study | OP951089 | OP945756 |
| 150930 | *Halopsis ocellata* | *Lafoeina tenuis* | P | Haltenbanken | 64.430 N 8.826 E | 201 | Polychaete | This study | OP951090 | – |
| 150931 | *Halopsis ocellata* | *Lafoeina tenuis* | P | Korsfjord | 60.151 N 5.113 E | 680 | Polychaete | This study | OQ031447 | – |
| 150932 | *Halopsis ocellata* | *Lafoeina tenuis* | P | Fedje | 60.749 N 4.480 E | 381 | Hydroid | This study | OP951091 | – |
| 150933 | *Halopsis ocellata* | *Halopsis ocellata* | M | Raunefjord | 60.257 N 5.139 E | 250–0 | NA | This study | OQ031439 | OQ031460 |
| 150934 | *Halopsis ocellata* | *Halopsis ocellata* | M | Korsfjord | 60.184 N 5.195 E | 670–0 | NA | This study | OQ031441 | – |
| 150935 | *Halopsis ocellata* | *Halopsis ocellata* | M | Ny Ålesund | 78.920 N 12.18 E | 94–0 | NA | This study | OQ031443 | OQ031464 |
| 150936 | *Halopsis ocellata* | *Halopsis ocellata* | M | Korsfjord | 60.151 N 5.099 E | 610–0 | NA | This study | OQ031442 | OQ031463 |
| | *Halopsis ocellata* | *Halopsis ocellata* | M | Raunefjord | 60.274 N 5.202 E | 20–0 | NA | *Schuchert, Hosia & Leclére (2017)* | KY363947 | MF000506 |
| | *Mitrocomella polydiademata* | *Lafoeina tenuis* | P | Flatevossen | 60.268 N 5.208 E | 30 | Hard substrate | This study | OQ031446 | OQ031467 |
| 150937 | *Mitrocomella polydiademata* | *Lafoeina tenuis* | P | Tvibyrge | 61.338 N 4.853 E | 38 | *Halecium* sp. | This study | OQ031431 | OQ031467 |
| 150938 | *Mitrocomella polydiademata* | *Lafoeina tenuis* | P | Gavlodden | 67.225 N 14.70 E | 33 | *Abietinaria* sp. | This study | OQ031435 | OQ031451 |
| 150939 | *Mitrocomella polydiademata* | *Lafoeina tenuis* | P | Tvibyrge | 61.338 N 4.853 E | 38 | *Sertularella* sp. | This study | OQ031434 | OQ031455 |
| | *Mitrocomella polydiademata* | *M. polydiademata* | M | Flatevossen | 60.268 N 5.208 E | 30–0 | NA | This study | OQ031450 | OQ031454 |
| 150940 | *Mitrocomella polydiademata* | *M. polydiademata* | M | North Sea | 57 N 3.65 E | 53–0 | NA | This study | OP951092 | – |
| 150941 | *Mitrocomella polydiademata* | *M. polydiademata* | M | Fanafjord | 60.247 N 5.286 E | 150–0 | NA | This study | OP951093 | – |
| 150942 | *Mitrocomella polydiademata* | *M. polydiademata* | M | North Sea | 57 N 3.65 E | 53–0 | NA | This study | OQ031437 | OQ031457 |
| 150943 | *Mitrocomella polydiademata* | *M. polydiademata* | M | Skagerrak | 58.882 N 9.685 E | 38–0 | NA | This study | OQ031432 | OQ031452 |
| | *Mitrocomella polydiademata* | *M. polydiademata* | M | Skagerrak | 58.634 N 10.26 E | 292–274 | NA | This study | OQ031445 | OQ031466 |
| 150944 | *Mitrocomella polydiademata* | *M. polydiademata* | M | Skagerrak | 58.882 N 9.685 E | 38–0 | NA | This study | OQ031448 | OQ031468 |

| Catalogue number (ZMBN) | Definitive ID | Initial ID | LS | Location | Lat/Long | Depth (m) | Substrate | Source | 16S | COI |
|---|---|---|---|---|---|---|---|---|---|---|
| | *Mitrocomella polydiademata* | *M. polydiademata* | M | ND | ND | ND | NA | *Kayal et al. (2015)* | KU710349 | – |
| | *Mitrocomella polydiademata* | *M. polydiademata* | M | Fanafjord | 60.240 N 5.229 E | 20–0 | NA | *Schuchert, Hosia & Leclére (2017)* | KY363949 | MF000508 |
| | *Mitrocomella polydiademata* | *M. polydiademata* | M | Scotland | 56.455 N 5.434 W | 0 | NA | *Schuchert, Hosia & Leclére (2017)* | KY363939 | MF000501 |
| | *Mitrocomella polydiademata* | *M. polydiademata* | M | Canada | 58.856 N 94.23 W | ND | NA | GenBank | – | MG423333 |
| | *Lafoeina* sp. | *Lafoeina tenuis* | P | Azores | ND | 400–340 | ND | *Moura et al. (2012)* | JN714673 | – |
| | *Earleria panicula* | *Lafoeina tenuis* | P | Sweden | 58.693 N 11.04 E | 130 | ND | BOLD | – | SWEMA1022-15 |
| | *Earleria panicula* | *Campanulina panicula* | P | Norway | ND | ND | ND | *Leclère et al. (2009)* | FJ550511 | – |
| | *Cosmetira pilosella* | *C. pilosella* | M | Norway | 60.184 N 5.196 E | 600–0 | NA | *Schuchert, Hosia & Leclére (2017)* | KY363955 | – |
| 150945 | *Cosmetira pilosella* | *C. pilosella* | M | Norway | 60.184 N 5.196 E | 670–0 | NA | This study | OQ031444 | OQ031465 |
| 150946 | *Cosmetira pilosella* | *C. pilosella* | M | Norway | 60.184 N 5.196 E | 670–0 | NA | This study | OQ031436 | OQ031456 |
| | *Cyclocanna producta* | *C. producta* | P | Norway | 58.755 N 9.656 E | 240 | Hydroid | This study | OQ031433 | OQ031453 |
| | *Cyclocanna producta* | *C. producta* | M | Norway | 60.184 N 5.195 E | 670–0 | NA | *Schuchert, Hosia & Leclére (2017)* | KY570308 | KY570317 |
| 150947 | *Cyclocanna producta* | *C. producta* | M | Norway | 60.257 N 5.139 E | 250–0 | NA | This study | OQ031449 | OQ031469 |
| 150948 | *Cyclocanna producta* | *C. producta* | M | Norway | 60.184 N 5.195 E | 670–0 | NA | This study | OQ031438 | OQ031459 |
| 150949 | *Earleria panicula* | *Earleria panicula* | M | Norway | 60.184 N 5.195 E | 670–0 | NA | *Schuchert, Hosia & Leclére (2017)* | KY570303 | KY570312 |
| 150950 | *Earleria panicula* | *Earleria panicula* | M | Norway | 60.184 N 5.195 E | 670–0 | NA | *Schuchert, Hosia & Leclére (2017)* | KY570306 | KY570315 |
| 150951 | *Earleria panicula* | *Earleria panicula* | M | Norway | 60.257 N 5.139 E | 250–0 | NA | *Schuchert, Hosia & Leclére (2017)* | KY570307 | KY570316 |
| 150952 | *Earleria panicula* | *Earleria panicula* | P | Norway | 60.162 N 5.176 E | 90 | Porifera | This study | OQ031440 | OQ031461 |

positive bands were then purified with ExoSAP-IT (Thermo Fisher Scientific, Inc., Waltham, MA, USA).

The resulting sequences were blasted against the nucleotide database of the National Centre for Biotechnology Information (NCBI, Bethesda, MD, USA) to check for apparent contaminations, and their chromatograms were visually checked in FinchTV (Geospiza, Inc., Denver, CO, USA) for weak or erroneous bases, which were then replaced using the nucleotide ambiguity code. The software Geneious v.11.1.5 (https://www.geneious.com) was used for contig assembly and generation of consensus sequences.

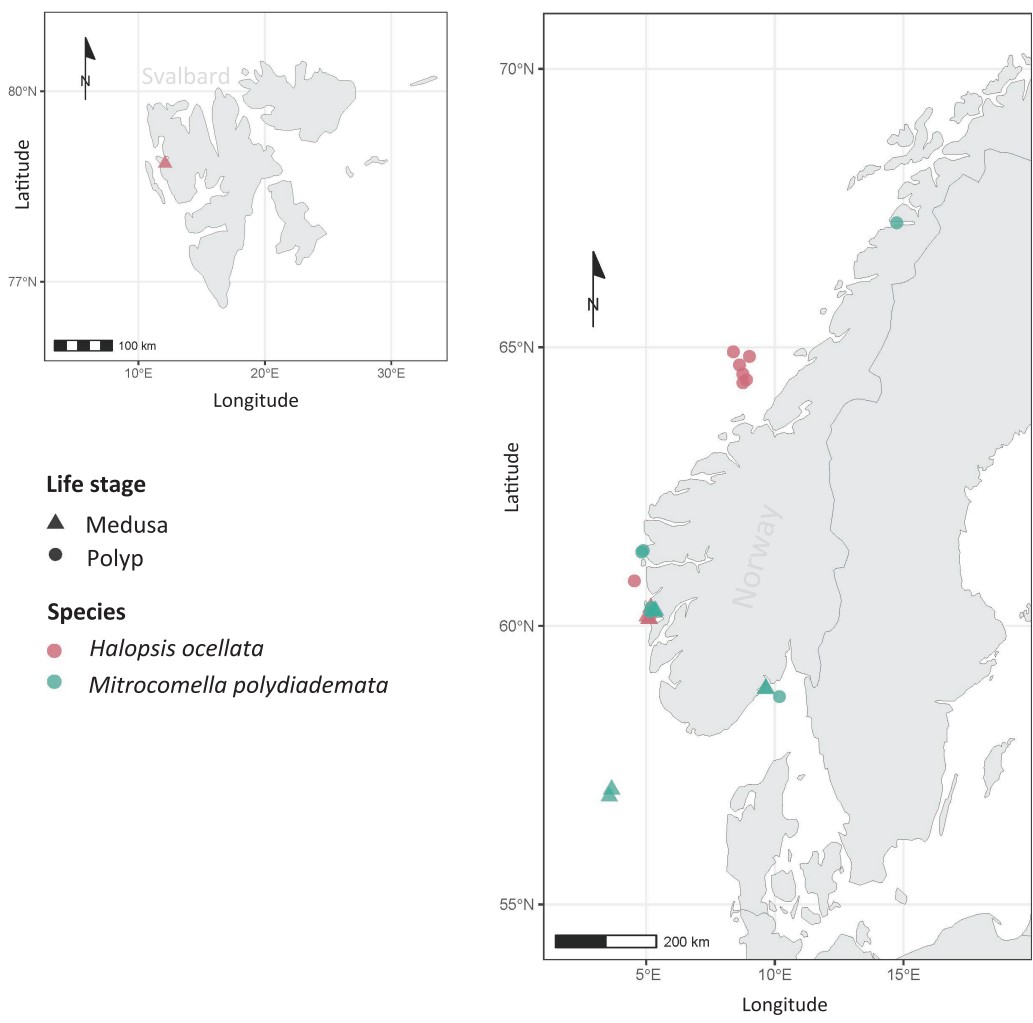

**Figure 1 Sampling localities.** Sampling localities for *Mitrocomella polydiademata* (green) and *Halopsis ocellata* (red), including both medusa (triangles) and polyp stages (circles). All polyp colonies were initially identified as *Lafoeina tenuis* based on morphology, but were later re-allocated to their corresponding medusa-based species through the phylogenetic and species delimitation analyses.

## Phylogenetic analyses

All available COI and 16S sequences labelled as *Lafoeina tenuis*, *Mitrocomella polydiademata*, and *Halopsis ocellata* were mined from the DNA Barcode of Life Database (BOLD, www.boldsystems.org; *Ratnasingham & Hebert, 2007*) and NCBI GenBank as well as from the UMB hydrozoan database (see Table 1 for a complete overview of the included sequences). This resulted in a final dataset of 32 COI sequences with 658 bases, and 40 16S sequences with 599 bases. As putative and potential outgroups, the following leptothecate taxa were used: for the 16S dataset *Hebella venusta* (*Allman, 1877*) and *Halisiphonia arctica* Kramp, 1932; for the COI dataset *Modeeria rotunda* (*Quoy & Gaimard, 1827*); and for both the 16S and COI datasets *Ptychogena crocea Kramp & Damas, 1925*, *Ptychogena lactea Agassiz, 1865* and *Staurostoma mertensii* (*Brandt, 1834*). Additionally, several novel sequences for Norwegian specimens of the mitrocomid species *Cyclocanna producta* (*Sars,*

**Table 2 The PCR specifications used for DNA barcoding.** PCR specifications for COI and 16S. S = number of sites in bp.

| Region | Forward primer | Reverse primer | S | Source | PCR settings |
|---|---|---|---|---|---|
| COI | LCO-1490 5′-GGTCAACAAATCATAAAGATATTGG-3′ | HCO-2198 5′-TAAACTTCAGGGTGACCAAAAAATCA-3′ | 650 | *Folmer et al. (1994)* | a. 94 °C for 5 min. b. 94 °C for 45 s. c. 45 °C for 30 s. d. 72 °C for 1 min. e. Go to b. and repeat four times f. 94 °C for 45 s. g. 50 °C for 30 s. h. 72 °C for 1 min. i. Go to f. and repeat 30 times j. 72 °C for 10 min |
| 16S | SHA 5′ACGGAATGAACTCAAATCATGT-3′ | SHB 5′-TCGACTGTTTACCAAAAACAT-3′ | 600 | *Cunningham & Buss (1993)* | a. 94 °C for 5 min. b. 94 °C for 30 s. c. 50 °C for 30 s. d. 72 °C for 1 min. e. Go to b. and repeat 39 times f. 72 °C for 7 min |

*1874*), *Earleria panicula* (*Sars, 1874*), and *Cosmetira pilosella Forbes, 1848* were included in the analysis. These potential outgroups and additional taxa were selected based on their phylogenetic position close to Mitrocomidae and Campanulinidae as shown by *Maronna et al. (2016)*.

For each marker, the selected sequences were used to construct a multiple alignment with the program MUSCLE (*Edgar, 2004*) as implemented in AliView v.1.26 (*Larsson, 2014*). Both alignments were then checked for obvious errors and their ends were trimmed by a few bases to eliminate poorly aligned flanking regions and achieve similar lengths. For the COI dataset, sequences were additionally translated using the minimally derived genetic code (Mold, Protozoan, and Coelenterate Mitochondrial Code) to check for the presence of stop codons (TAA and TAG for cnidarians). Each alignment was processed separately during phylogenetic reconstruction with a maximum likelihood (ML) approach. All ML analyses were performed using the web server W-IQ-TREE (*Trifinopoulos et al., 2016*, http://iqtree.cibiv.univie.ac.at/). The substitution models were estimated with the implemented modeltest option and a set FreeRate heterogeneity. The best score models were TIM3+F+G4 and TIM2+F+I for 16S and COI, respectively. The analyses were subsequently run with 5,000 repetitions and the ultrafast bootstrap option (*Hoang et al., 2017*).

**Molecular species delimitation**

Molecular species delimitation was performed using two different methods: the Automatic Barcode Gap Discovery (ABGD; *Puillandre et al., 2011*) and the Bayesian Poisson tree process model (bPTP; *Zhang et al., 2013*). For both analyses the respective online servers were used with default settings. Reduced alignments (*i.e.*, including all sequences with the

exception of the outgroups) for both 16S and COI were used as input for ABGD. The ML trees were used as input for bPTP. In addition, pairwise distances were calculated for both alignments using the K80 (or Kimura's two-parameter model) with 500 bootstrap replicates in MEGA version 10.2.5 (*Kumar et al., 2018*) in order to estimate inter- and intra-specific genetic distances. The distances were calculated based on the species hypothesis recovered in the phylogenetic and species delimitation analyses.

### Morphological analysis

To characterize the putative species recovered in the phylogenetic and species delimitation analyses, the following eight morphological characters were measured from each hydroid specimen: hydrothecal length, hydrothecal width, nematothecal length, nematothecal width, length and width of undischarged isorhiza capsules, and length and width of undischarged mastigophore capsules. Five replicate measurements of each character were performed per specimen.

Statistical analyses were carried out in RStudio version 1.4.1106 (*R Core Team, 2017*) to compare the measurements of morphological characters between putative species and to test for significant differences. Since the data were non-parametric and non-homogeneous, a series of Wilcoxon rank sum tests was performed. To explore the variation within specimens, all five measurements per specimen were plotted using the ggplot2 package (*Wickham, 2016*). A principal component analysis (PCA) was then used to characterize the distribution of the measurements along axes of major variation. For this PCA, all missing data were removed from the morphological dataset and the mean values of the measurements were taken for each specimen. The PCA was generated with the packages factoextra and FactoMineR (*Sebastien, Josse & Husson, 2008*), using the PCA() function which enables automatic scaling of the units. A Scree plot (Fig. S1) was used to check that the first two dimensions explain >75% of the variation. For visualization, a biplot was created using the fviz_pca_biplot() function (Fig. S2).

## RESULTS

A total of 23 specimens belonging to the target taxa were examined in this work. Four of them were morphologically identified as hydromedusae of *Halopsis ocellata*, seven as hydromedusae of *Mitrocomella polydiademata*, and twelve were polyp colonies morphologically referable to *Lafoeina tenuis*. The literature used for morphological identification included original species descriptions and later taxonomic studies (*Agassiz, 1865*; *Sars, 1874*; *Romanes, 1876*; *Russell, 1953*; *Edwards, 1973a*; *Cornelius, 1995a*). The sampling localities for all specimens are shown in Fig. 1.

### Phylogenetic analysis and species delimitation

The phylogenetic and species delimitation analyses revealed a mismatch between the genetic identity of the specimens and their morphological species identification. Both 16S and COI trees resulted in one distinct clade with high support values for each of the mitrocomid species *Halopsis ocellata* and *Mitrocomella polydiademata*, but each of these clades also included sequences of *Lafoeina tenuis* which were genetically identical to those

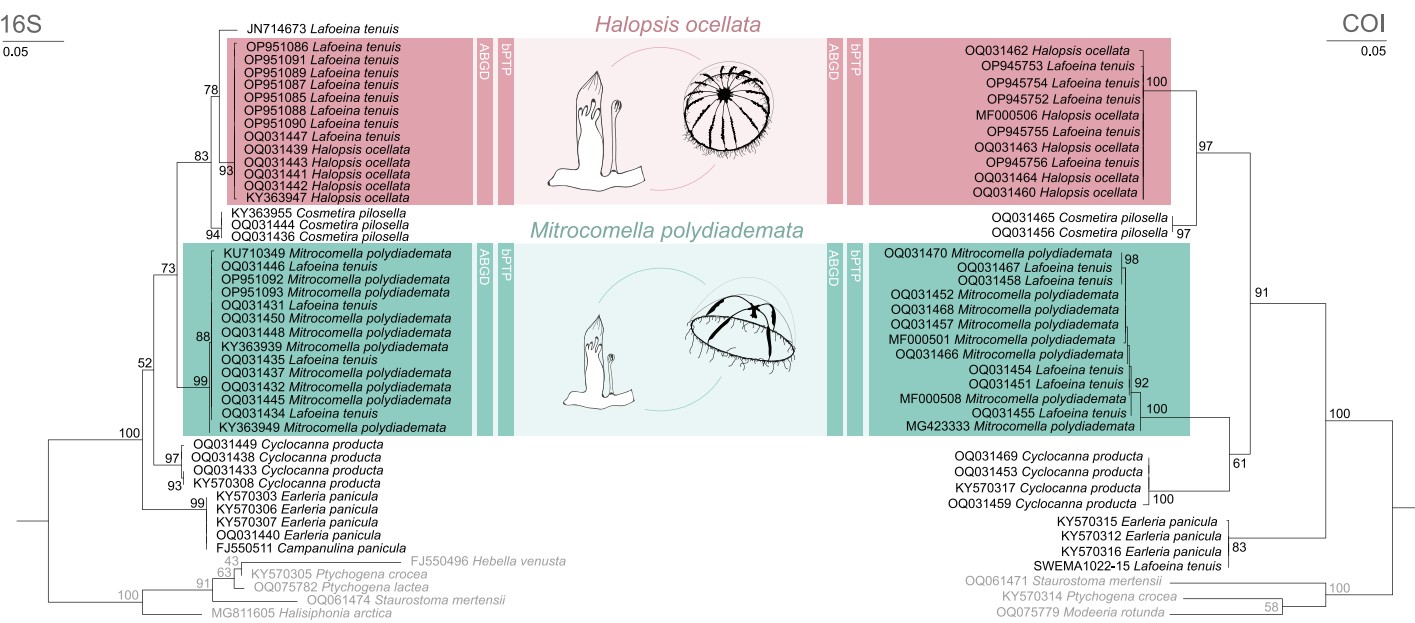

**Figure 2 Maximum likelihood phylogenies for 16S and COI.** Maximum likelihood phylogenies of 16S (left) and COI (right) for the analysed specimens. Bars indicate delimitation results based on ABGD and bPTP. In grey the outgroup taxa. Support values are bootstrap values.

obtained from the respective hydromedusae (Fig. 2). Moreover, two additional *L. tenuis* sequences were recovered outside of these clades. COI sequence SWEMA1022-15 from the BOLD database grouped together with several sequences of *Earleria panicula*. A closer inspection of the associated voucher image in the BOLD database revealed that the specimen belongs to *E. panicula* as suggested by the phylogenetic placement of its COI sequence. The sequence is thus considered misidentified and is excluded from further discussion. The 16S sequence JN714673 from GenBank is placed separately from both *H. ocellata* and *M. polydiademata* and likely represents a distinct independent clade in the 16S tree. This sequence is closely related to but genetically distinct from the *Halopsis ocellata* clade. All clades identified in the phylogenetic analyses were recovered as putative species by the molecular species delimitation methods, confirming that *H. ocellata* forms a cohesive unit with some *L. tenuis* polyps, while other *L. tenuis* polyps correspond to the polyp stage of *M. polydiademata* (Fig. 2).

## Genetic distances

Genetic distances between identified clades (between species) and intra-clade (within species) are presented in Table 3 (16S) and Table 4 (COI) for the target taxa plus additional mitrocomid species. For 16S (Table 3), mean intraspecific distances (±SE) ranged from 0.00 ± 0.00% in *E. panicula* and *C. pilosella* to 0.2 ± 0.13% in *C. producta*, with an overall mean of 0.06 ± 0.04%. Conversely, mean interspecific distances ranged from 2.5 ± 0.7% (between *H. ocellata*, *L. tenuis* JN714673, and *C. pilosella*) to 8.4 ± 1.3% (between *H. ocellata* and *E. panicula*), with an overall mean of 5.8 ± 1.0%. There is a distinct gap in interspecific distances of ca. 5.5% difference between *H. ocellata* and *M. polydiademata*,

**Table 3 Genetic distances for 16S.** Mean K80 pairwise genetic distances within clades (in bold, % ± standard error) and between clades (% ± standard error) for 16S.

| | H. ocellata | M. polydiademata | E. panicula | L. tenuis (JN714673) | C. pilosella | C. producta |
|---|---|---|---|---|---|---|
| *Halopsis ocellata* | **0.05 ± 0.04** | | | | | |
| *Mitrocomella polydiademata* | 5.5 ± 0.9 | **0.05 ± 0.03** | | | | |
| *Earleria panicula* | 8.4 ± 1.3 | 8.1 ± 1.3 | **0** | | | |
| *Lafoeina tenuis* (JN714673) | 2.5 ± 0.7 | 5.5 ± 1.1 | 8.1 ± 1.3 | **–** | | |
| *Cosmetira pilosella* | 2.5 ± 0.7 | 5.1 ± 1 | 7.1 ± 1.2 | 2.5 ± 0.7 | **0** | |
| *Cyclocanna producta* | 6.5 ± 1.1 | 5.3 ± 1 | 6.1 ± 1 | 6.7 ± 1.2 | 6.3 ± 1.1 | **0.2 ± 0.13** |

**Table 4 Genetic distances for COI.** Mean K80 pairwise genetic distances within clades (in bold, % ± standard error) and between clades (% ± standard error) for COI.

| | H. ocellata | M. polydiademata | E. panicula | C. pilosella | C. producta |
|---|---|---|---|---|---|
| *Halopsis ocellata* | **0.1 ± 0.05** | | | | |
| *Mitrocomella polydiademata* | 13.2 ± 1.5 | **0.37 ± 0.13** | | | |
| *Earleria panicula* | 13.4 ± 1.4 | 15.2 ± 1.7 | **0.09 ± 0.09** | | |
| *Cosmetira pilosella* | 5.5 ± 0.9 | 11.7 ± 1.4 | 13.8 ± 1.6 | **0** | |
| *Cyclocanna producta* | 10.8 ± 1.3 | 10.2 ± 1.3 | 14 ± 1.6 | 12 ± 1.3 | **0.23 ± 0.13** |

indicating that they are separate species. Within both species there is less than 0.1% genetic difference between the sequences originated from hydromedusae and polyps.

For COI (Table 4) mean intra-species distances (±SE) ranged from 0.00 ± 0.00% in *C. pilosella* to 0.37 ± 0.13% in *M. polydiademata*, with an overall mean of 0.15 ± 0.08%. Mean interspecific distances ranged from 5.5 ± 0.9% (*H. ocellata* and *C. pilosella*) to 15.2 ± 1.7% (between *M. polydiademata* and *E. panicula*), with an overall mean of 11.9 ± 1.42%. There is a distinct gap in interspecific distances of ca. 13.2% difference between *H. ocellata* and *M. polydiademata*, indicating that they are separate species. The respective intraspecific distances are less than 0.4%.

## Morphological analyses and ecological preferences

The hydroid colonies in the *H. ocellata* and *M. polydiademata* clades differed from each other in hydrothecal, nematothecal, and nematocyst size, as well as substrate preference and depth distribution. The morphometric analyses suggest that the different morphological characters analyzed from the polyp stage of *M. polydiademata* are, on average, consistently smaller than those of *H. ocellata* (Table 5, Figs. 3–6). All statistical comparisons showed significant differences between *M. polydiademata* and *H. ocellata* polyps, but the range of most characters had at least some overlap between the two groups, preventing characterization of the polyps based solely on their morphological traits (Fig. 3). The analyses identified the length of undischarged mastigophore capsules as the most promising diagnostic morphological character, as the differences observed between

**Table 5 Statistical results for the morphological measurements for polyp characters of *Halopis ocellata* and *Mitrocomella polydiademata*.** Mean (±SD) values and Wilcoxon rank sum test W and *p* values of all evaluated morphological polyp characters for *Halopsis ocellata* (*n* = 6) and *Mitrocomella polydiademata* (*n* = 3). Five replicate measurements of each character were performed per specimen.

| Character | Mean ± SD (in µm) | | W | *p* value |
|---|---|---|---|---|
| | *H. ocellata* | *M. polydiademata* | | |
| Nematotheca length | 217 ± 131 | 93 ± 25 | 361 | 0.001105 |
| Nematotheca width | 26 ± 4 | 19 ± 2 | 424.5 | 1.653e−06 |
| Mastigophore length | 27 ± 3 | 16 ± 2 | 450 | 6.45e−08 |
| Mastigophore width | 4.5 ± 0.7 | 3 ± 0.6 | 421.5 | 2.351e−06 |
| Isorhiza length | 7 ± 1 | 5.7 ± 0.6 | 418.5 | 3.355e−06 |
| Isorhiza width | 2.4 ± 0.4 | 2.2 ± 0.3 | 323.5 | 0.01827 |
| Hydrotheca length | 350 ± 93 | 280 ± 58 | 344 | 0.004317 |
| Hydrotheca width | 125 ± 20 | 104 ± 11 | 370.5 | 0.0004772 |

colonies of the two species were highly significant and the measurements obtained did not show any degree of overlap.

The sampling events covered various locations in Norwegian waters for both polyps and medusae (Fig. 1). In this study, *H. ocellata* was collected from the Arctic Ocean (Svalbard) in the north to the North Sea (Bergen) in the south, while *M. polydiademata* was collected from the Norwegian Sea (Bodø), the North Sea, and the Skagerrak. Interestingly, the depth of collection for polyps differed between the two species. All polyp specimens assigned to *M. polydiademata* were collected at relatively shallow waters (≤40 m), whereas all *H. ocellata* polyps were collected from deeper sites between 187–680 m (Table 1). For *M. polydiademata* polyps, the substrate was other thecate hydroids ("macrocolonial" colonies such as *Sertularella polyzonias*, *Halecium* sp., *Abietinaria* sp.), algae and inorganic hard substrates, while *H. ocellata* was found growing mostly on polychaete tubes, but also sponges, and other unidentified thecate hydroids.

The analysis of morphological characters resulted in the following integrative descriptions for the polyp stages of *H. ocellata* and *M. polydiademata*:

*Halopsis ocellata* Agassiz, 1865
Figures 4A–4F
Material examined: ZMBN150926, ZMBN150927, ZMBN150928, ZMBN150929, ZMBN150930, ZMBN150931 (Table 1).

**Description.** Colony minute, '*Lafoeina tenuis*'-type, stolonal. Density and distribution of hydrothecae and nematothecae variable within one colony and between different colonies, at times both structures concentrated and numerous, other times relatively separated from each other and with few nematothecae; hydrothecae and nematothecae arising directly from hydrorhiza. Hydrorhiza branching, anastomosing or not, with smooth perisarc, without septae. Hydrotheca 215–500 µm high, 81–167 µm wide, tubular but slightly flared at origin of operculum, straight or slightly curving, sessile, separated from hydrorhiza by

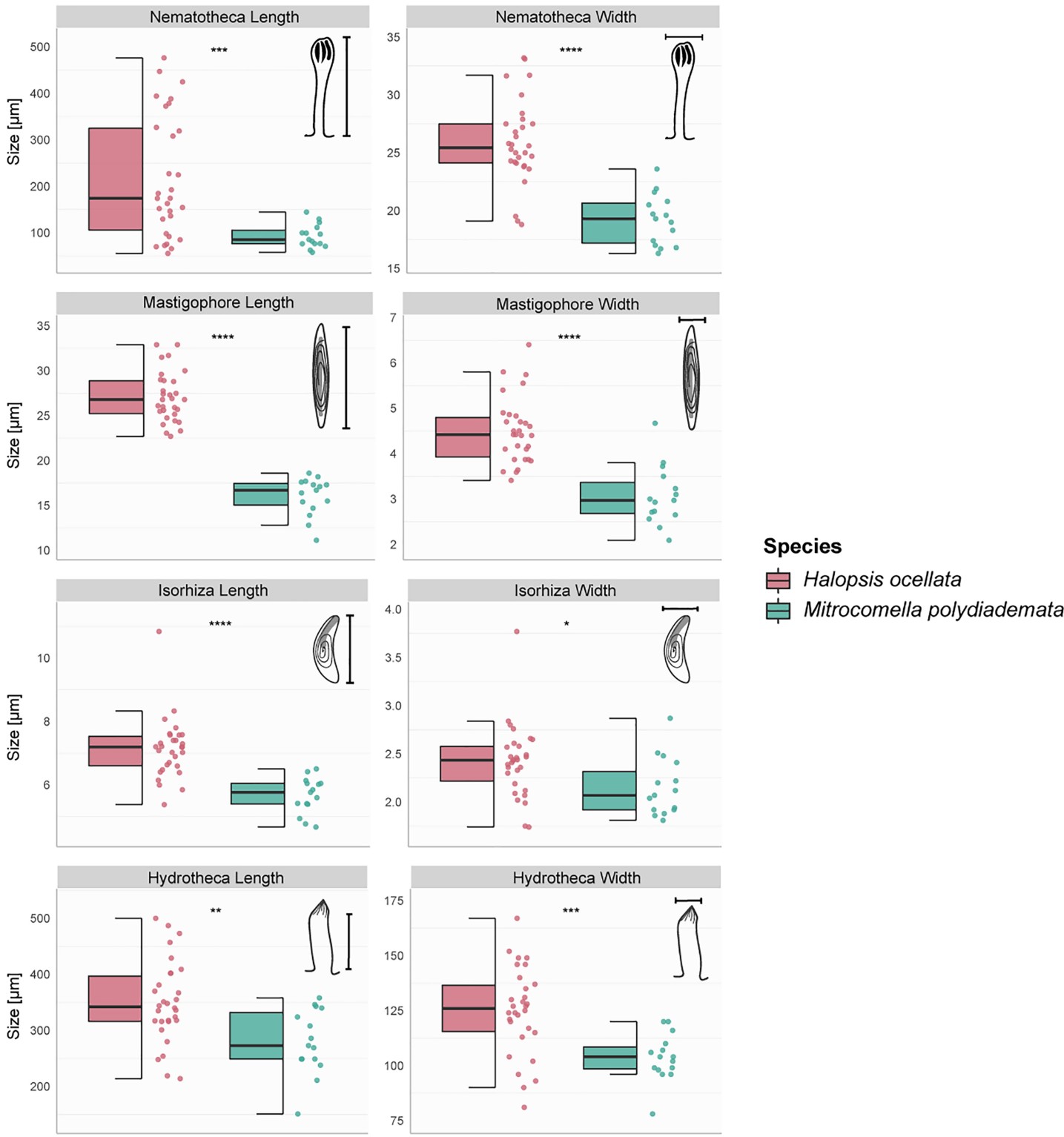

**Figure 3** **Boxplots showing the morphological measurements from *Halopsis ocellata* and *Mitrocomella polydiademata* polyps.** Variation in eight morphological characters from the polyp stage of *Halopsis ocellata* (red) and *Mitrocomella polydiademata* (green). Significance for the Wilcoxon test: ****$p < 0.0001$, ***$p < 0.001$, **$p < 0.01$, *$p < 0.05$.

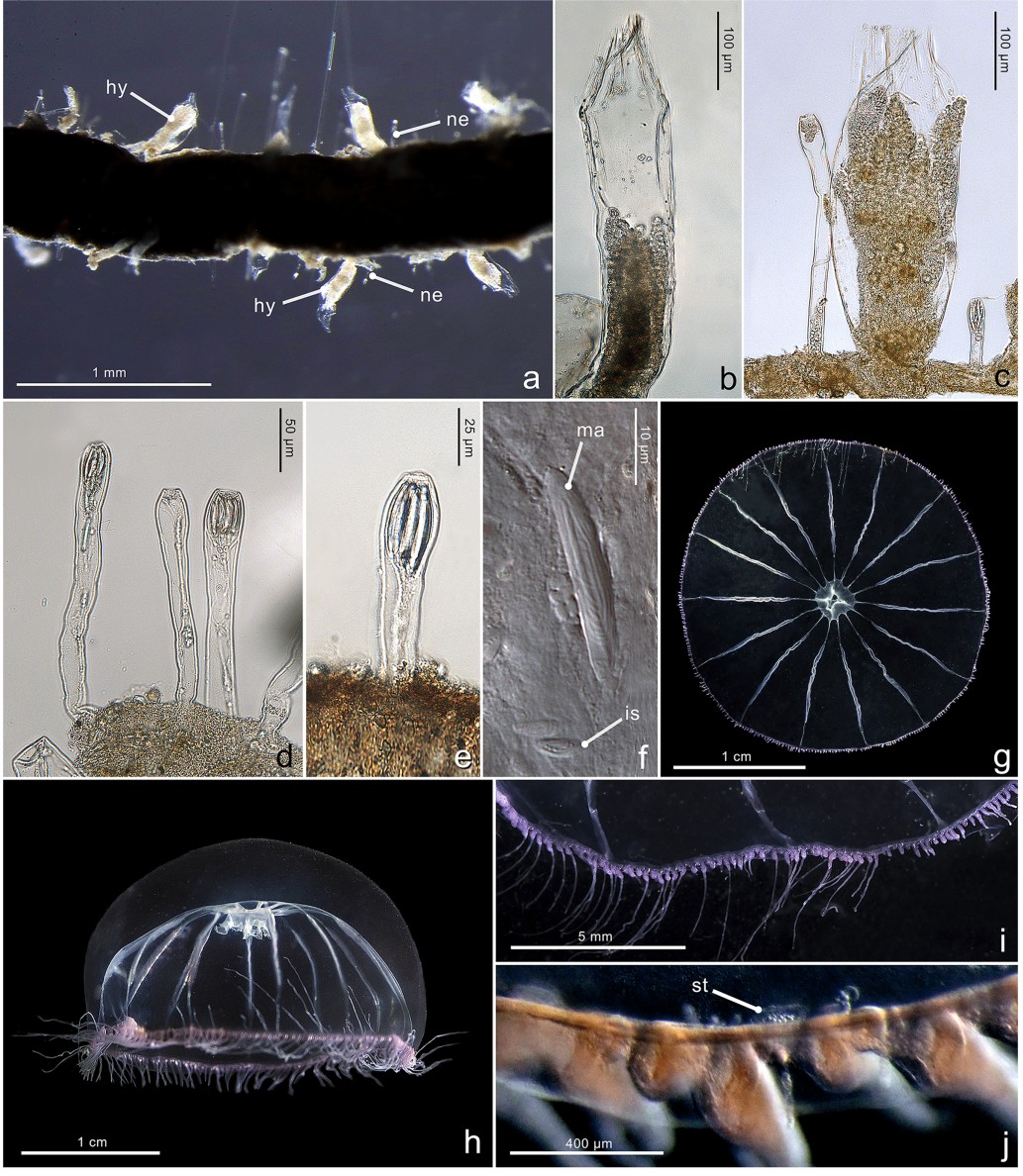

**Figure 4 Morphological characters of *Halopsis ocellata*.** *Halopsis ocellata*. (A) Colony growing on a polychaete tube. (B, C) Hydrothecae (hy) with polyps. (D, E) Nematothecae (ne). (F) Mastigophore (ma) and Isorhiza (is). (G, H) Adult hydromedusa stage. (I, J) Details of the umbrella margin of the adult hydromedusa including a statocyst (st). Image Credits: Lara M. Beckmann, Fredrik Broms (G–I), Joan J. Soto-Angel.

basal constriction or less frequently having no evident constriction. Operculum steep, consisting of a folded continuation of the hydrothecal walls, giving the appearance of 9–11 triangular pleats meeting centrally, without basal crease-line in preserved material. Hydranth with approx. 12 tentacles, without intertentacular web, with conical hypostome. Nematotheca 56–476 µm high, 19–33 µm wide, always pedicellate and mostly long and narrow, 1/3 to 1/1 as long as the hydrotheca, often slightly bulbous distally, arising individually throughout the colony, with distal opening. Gonotheca unknown, but gives

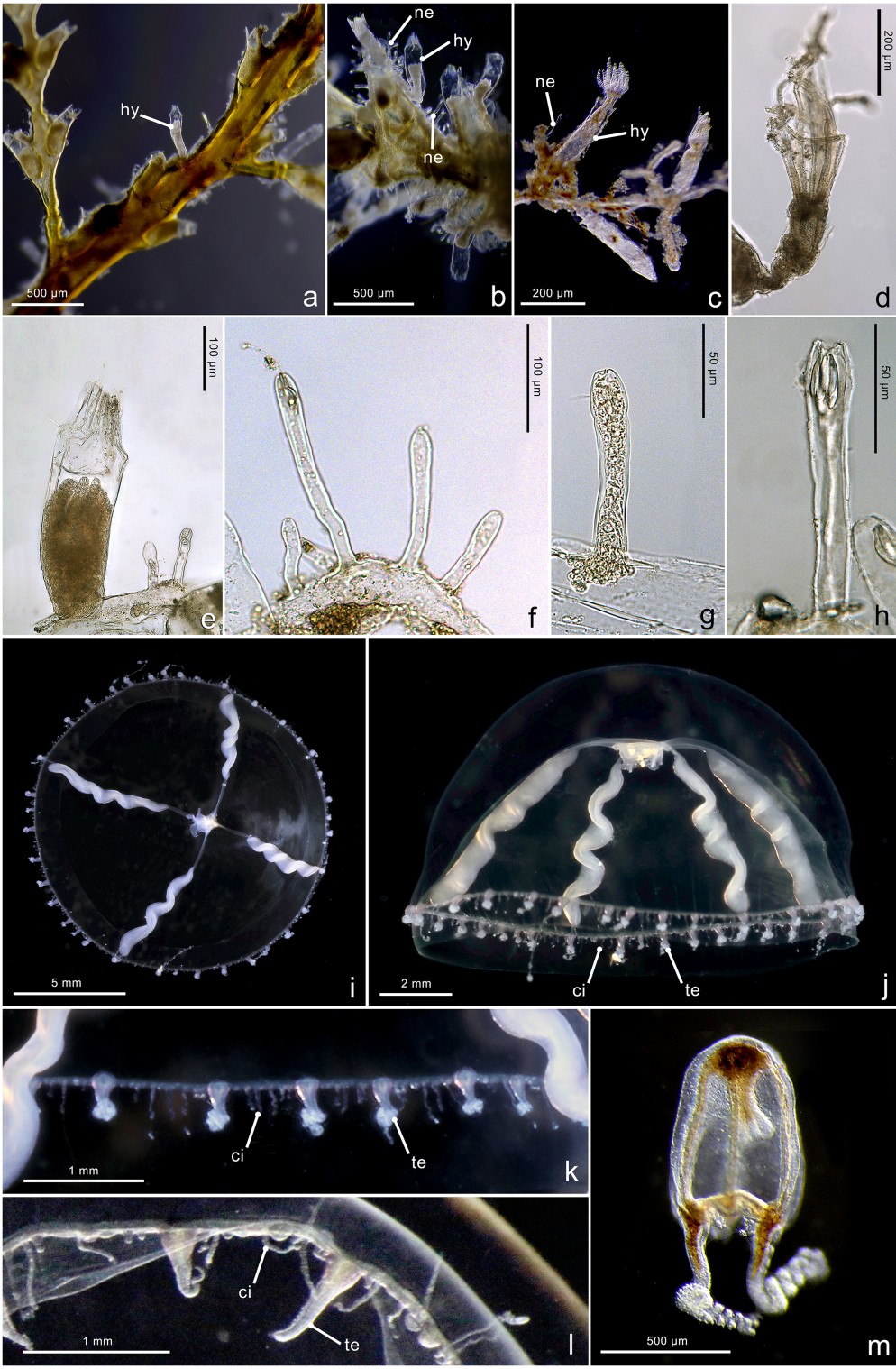

**Figure 5 Morphological characters of *Mitrocomella polydiademata*.** *Mitrocomella polydiademata*. (A–C) Colony growing on a large hydroid. (D) Polyp with extended tentacles. (E) Hydrotheca (hy) with polyp and two nematothecae (ne). (F–H) Nematothecae. (I, J) Adult hydromedusa stage. (K, L) Margin details of the adult hydromedusa with cirri (ci) and tentacles (te). (M) Newly released hydromedusa. Image Credits: Lara M. Beckmann, Joan J. Soto-Angel.

rise to medusae conforming to the species *Halopsis ocellata*. Two types of nematocysts identified: mastigophores (length 23–33 μm, width 3.4–6.4 μm) and isorhizas (length 5.4–10.8 μm, width 1.7–3.8 μm).

**Substrate.** Growing on various substrates, mostly on polychaete tubes, but also on the hydrocaulus of unidentified thecate hydroid colonies, and sponges.

**Depth range**. Waters below 180 m (polyps in this study were collected from 180–680 m*)*

***Mitrocomella polydiademata*** (*Romanes, 1876*).
    Figures 5A–5H
    Material examined: ZMBN150937, ZMBN150938, ZMBN150939 (Table 1).

**Description**. Colony minute, '*Lafoeina tenuis*'-type, stolonal. Hydrothecae and nematothecae variable in number and distribution, both inside each colony and between different colonies, always arising directly from hydrorhiza. Hydrorhiza branching, sometimes anastomosing, with smooth perisarc, lacking internal septa. Hydrotheca 151–358 μm high, 78–120 μm wide, tubular but slightly flared at origin of operculum, straight or slightly curving, sessile, most often separated from hydrorhiza by basal constriction but this not always evident. Operculum steep, a continuation of the hydrothecal wall which folds upon itself forming 8–12 pleats meeting centrally, without basal crease-line in either live or preserved material. Hydranth extensile, at least two times the length of the hydrotheca when extended, with 10–12 amphicoronate tentacles, without intertentacular web, with conical hypostome. Nematotheca 58–144 μm high, 16–24 μm wide, always pedicellate, 1/3 as long as the hydrotheca, often slightly bulbous distally, arising individually throughout the colony, with distal opening. Gonotheca unknown, but gives rise to medusae conforming to the species *Mitrocomella polydiademata*. Two types of nematocysts identified: mastigophores (length 11.1–18.6 μm, width 2.1–4.7 μm) and isorhizas (length 4.7–6.5 μm, width 1.8–2.9 μm).

**Substrate**. Growing on various substrates, including the hydrocaulus of other thecate hydroids (*Sertularella polyzonias*, *Halcium* sp., *Abietinaria* sp.), red algae, and inorganic hard substrates.

**Depth range.** Waters above 50 m (polyps in this study were collected from 30–38 m).

## DISCUSSION

Hydroid colonies morphologically referable to *Lafoeina tenuis* constitute the polyp stage of at least two different species of hydromedusae: *Halopsis ocellata* and *Mitrocomella polydiademata*. This finding, confirmed by the phylogenetic and molecular delimitation analyses of both COI and 16S markers, results in a series of taxonomic and biogeographic implications that challenge our current understanding of hydrozoan diversity in the North Atlantic.

We present the first description of the previously unknown polyp stage of *H. ocellata*, as well as the only hitherto known hydroid in genus *Halopsis*. A large and distinctive mitrocomid hydromedusa, *H. ocellata* has a mostly disjunct distribution in temperate-cold

regions, except for one record from the tropical Pacific (*Cely & Chiquillo, 1993*) that requires confirmation. It occurs in the North Atlantic and Arctic Oceans, as well as in subantarctic waters in the southwestern Atlantic and southeastern Pacific Oceans (*Kramp, 1961*; *Zelickman, 1972*; *Genzano, Mianzan & Bouillon, 2008*; *Oliveira et al., 2016*). Many of its records come from moderately deep or oceanic waters, particularly when these occur near the coast as in the Norwegian and Chilean fjords (*Russell, 1953*; *Cornelius, 1995a*; *Palma et al., 2014*). While it has long been assumed that there is a polyp stage in the life cycle of *H. ocellata* (*e.g.*, *Russell, 1953*), few attempts have been made to discover its identity. This is probably due to difficulties in rearing the jellyfish in laboratory conditions and the fact that, although relatively common, the medusae are never caught in large enough numbers to allow for extensive experimentation (*Russell, 1953*; *Cornelius, 1995a*). *Hadzi (1917)* speculated that *Campanulina hincksii* Hartlaub, 1897 could be the polyp of *H. ocellata*, but this hydroid has since been shown to produce hydromedusae referable to *Eucheilota maculata* *Hartlaub, 1894* (*Werner, 1968*), and no other candidate polyp species has been seriously considered as a potential match for *H. ocellata* until now. By describing the polyp stage of a *Halopsis* species, we reduce the number of genera in Mitrocomidae for which no information exists regarding benthic stages to only one (genus *Cosmetirella Browne, 1910*).

The uncovered link between *M. polydiademata* and *L. tenuis* reveals an incorrect or incomplete description of the polyp stage in the past. The polyp stage of this species was previously described by *Edwards (1973a)* as *Cuspidella*-like based on laboratory-reared cultures obtained from medusae collected in the wild. The hydroids of '*Cuspidella*'-type and '*L. tenuis*'-type are morphologically similar, as they differ only in the presence of nematothecae in the latter, and since these structures are sometimes minute and inconspicuous the two nominal taxa can easily be confused (*Bouillon, 1971*; *Migotto & Cabral, 2005*). Edwards, however, was a careful observer and an experienced hydrozoologist (*Edwards, 1964*, *1965*, *1973a*, *1973b*, *1978*, *1983*), and it is unlikely he would have missed the presence of nematothecae had these occurred in his specimens. Subsequent attempts to culture *M. polydiademata* seemed to partially confirm his results by obtaining young *Cuspidella*-type primary polyps (*Martin, Chia & Koss, 1983*; *Freeman & Ridgway, 1990*), but a degree of uncertainty persists because fully grown colonies have not been observed so far under laboratory conditions. Edwards collected his medusae on the west coast of Scotland, not far from the type locality of *M. polydiademata*. The morphology of those medusae matches the original description for the species as well as that of the specimens analyzed here. Consequently, it is unlikely that the different morphologies of the polyp stages for *M. polydiademata* could have stemmed from a mistaken identity. All known sequences obtained from Scottish and Canadian specimens form a monophyletic clade with the Norwegian specimens, and the species delimitation analyses also support a single species occurring in the entire North-Atlantic Ocean, further suggesting that the differences observed are not likely due to misidentification or the presence of cryptic species.

The lack of nematothecae in laboratory-reared colonies of *M. polydiademata* could instead be explained as a result of normal ontogenetic development and/or

environmentally-induced variation. The colonies of *M. polydiademata* reared by Edwards remained immature and never developed gonophores (*Edwards, 1973a*), and all subsequent attempts at rearing polyps from individual medusae have also failed at obtaining mature colonies (*Martin, Chia & Koss, 1983*; *Freeman & Ridgway, 1990*). As the development of *L. tenuis* from planula to reproductive colonies has never been observed, young *L. tenuis* colonies could go through a *Cuspidella*-type stage in which nematothecae are not yet developed. Non-reproductive *L. tenuis* colonies with fully-developed nematothecae are nonetheless commonly encountered in the field (*Sars, 1874*; *Vervoort, 2006*; *Moura, 2015*), suggesting no correlation between the presence of nematothecae and gonothecae. Environmental variability, on the other hand, impacts hydroid morphology and results in polymorphism within colonies (*e.g.*, *Dudgeon & Buss, 1996*; *Griffith & Newberry, 2008*; *Prudkovsky, Ekimova & Neretina, 2019*). Some characters in the polyp may not be expressed when the colonies are grown in the laboratory (*Brinckmann-Voss, 1970*; *Migotto, 1996*; *Migotto & Cabral, 2005*). Nematothecae, for example, did not arise in cultured colonies of *Cirholovenia tetranema* *Kramp, 1959* obtained from planuloids, and they were scarce or absent in some sections of the hydrorhiza in colonies tied in glass slides (*Migotto & Cabral, 2005*). Other structures, such as the filiform tentacles of *Stauridiosarsia ophiogaster* (*Haeckel, 1879*) and *Stauridiosarsia producta* (*Wright, 1858*), are absent from field-collected animals but develop in polyps kept in the laboratory (*Brinckmann-Voss, 1970*; *Migotto, 1996*).

Unexpected as it is, the link between *L. tenuis* polyps and *H. ocellata* and *M. polydiademata* medusae is not entirely surprising, as all mitrocomid jellyfish for which the polyp stage is known are produced by campanulinid hydroids. The first reference to a campanulinid polyp stage in Mitrocomidae was made as early as 1886, when the polyps of *Mitrocoma annae* *Haeckel, 1864* were described and morphologically attributed to genus *Cuspidella* *Hincks, 1866* (*Metchnikoff, 1886*), a taxon of questionable validity since it encompasses a collection of polyps belonging to various leptothecate hydromedusae referable to different unrelated families (*Cornelius, 1995a*). Metchnikoff further suggested that other taxa within Mitrocomidae might develop similar campanulinid hydroids, which was proven true when the *Cuspidella*-type polyps of *Cosmetira pilosella* *Forbes, 1848*, *Mitrocomella brownei* (*Kramp, 1930*), *Earleria purpurea* (*Foerster, 1923*), and *Mitrocoma cellularia* (*Agassiz, 1862*) were described (*Rees & Russell, 1937*; *Rees, 1941*; *Widmer, 2004*; *Widmer, 2011*). The polyp stage of the rest of the mitrocomid species with known life cycle is also a campanulinid, albeit not a *Cuspidella*: *Earleria corachloeae* *Widmer, Cailliet & Geller, 2010*, *Earleria panicula* (*Sars, 1874*), and *Cyclocanna producta* (*Sars, 1874*) possess a *Campanulina*-type, *Racemoramus*-type, and *Egmundella*-type polyp stage, respectively (*Widmer, Cailliet & Geller, 2010*; *Schuchert, Hosia & Leclére, 2017*).

Although morphological identification of *L. tenuis*-type polyps is currently unattainable, at least for the populations in Norwegian waters we have identified several morphological and ecological characters that consistently separate the polyps of *H. ocellata* from those of *M. polydiademata*. Too little is known of the variability of these characters in other campanulinid polyps to use them for reliable identification outside of our study area (*Calder, 1991*; *Cornelius, 1995a*; *Migotto & Cabral, 2005*), but our results provide a starting

set of hypotheses that will allow researchers to test for morphological differences in small, inconspicuous *L. tenuis*-type hydroids from other regions. Besides genetic identification, the polyp stages of *H. ocellata* and *M. polydiademata* were distinguished by a combination of their preferred habitat and the size of their hydrothecae, nematothecae, and mastigophore capsules, even if the analyzed characters (except for mastigophore length) overlap between the two taxa. In addition, vertical distribution was a good predictor of the species in our dataset, as the polyp stages of these two species apparently occupy different ecological niches in the region. Taken together, these characters allowed us to characterize *M. polydiademata* as a smaller and shallower species with shorter mastigophore capsules and comparatively small hydrothecae and nematothecae, and predominantly found in shallow waters <50 m in depth. In contrast, *H. ocellata* is larger, has longer mastigophores capsules, comparatively larger hydrothecae and nematothecae, and occurs in waters deeper than 150 m. Finally, while not as straightforward as the other characters, substrate preference offers an alternative clue for species identity in Norwegian waters, as *H. ocellata* polyps were more frequently encountered in association with polychaete tubes whereas *M. polydiademata* polyps were mostly found growing on other thecate hydroids. The size of hydrothecae, nematothecae, and nematocysts are all characters used in the diagnosis of other taxa within Hydrozoa (*Cornelius, 1995a*, *1995b*; *Schuchert, 2012*), but a thorough evaluation of their usefulness in distinguishing campanulinid and mitrocomid taxa is still lacking.

The existence of at least one 16S sequence from a *L. tenuis* specimen that does not cluster with either of the two analyzed hydromedusan species suggests that, besides *H. ocellata* and *M. polydiademata*, other hydromedusa-based taxa likely possess hydroids morphologically referable to *L. tenuis* in their life cycle. The geographic distribution of *L. tenuis* also does not match that of *H. ocellata*, *M. polydiademata*, or the combined distribution of these two hydromedusan taxa, supporting the status of the former as a species complex. *Lafoeina tenuis* is a widespread taxon with confirmed records from the northeastern and northwestern Atlantic Ocean (*Sars, 1874*; *Calder, 2003*; *Vervoort, 2006*; *Moura, 2015*), as well as the Arctic Ocean (*Ronowicz, Kukliński & Mapstone, 2015*); but the species has not been reliably observed in subantarctic waters, where *H. ocellata* occurs (*Kramp, 1961*; *Genzano, Mianzan & Bouillon, 2008*), or in the northwestern Pacific Ocean, where *M. polydiademata* occurs (*Arai & Brinkmann-Voss, 1980*; *Larson, 1987*; *Mills, 1993*). Conversely, *L. tenuis* is present in tropical and subtropical waters of the Gulf of Mexico and southern Atlantic (*Bouzon, Brandini & Rocha, 2012*; *Mendoza-Becerril, Simões & Genzano, 2018*), as well as in the Mediterranean Sea—the latter in the form of the synonymized name *Lafoeina vilaevelebiti Hadzi, 1917*—(*Isinibilir et al., 2015*; *Topcu et al., 2018*; *Yilmaz et al., 2020*), where no confirmed records of the two hydromedusan species exist. Disentangling the complicated taxonomic history of *L. tenuis* and testing the potential for cryptic diversity in *H. ocellata* and *M. polydiademata* requires a thorough taxonomic revision, an aim that is outside the scope of the present contribution. Until such a revision is made, we suggest that the nominal species *Lafoeina tenuis Sars, 1874* be considered a partial synonym of both *H. ocellata* and *M. polydiademata* and that polyps morphologically identified to *L. tenuis* be referred to as '*Lafoeina tenuis*'-type until further

associations are resolved. This temporary solution, while not optimal, is analogous to the one currently adopted for genus *Cuspidella* (*Cornelius, 1995a*), and has the benefit of incorporating the new findings in the status of these three species while minimizing the potential for a confusing situation in which the name of either hydromedusa is applied to unidentifiable polyps.

## CONCLUSIONS

Complete descriptions of life cycles are essential to correctly estimate the diversity of cnidarian species in an area, but for most campanulinid and mitrocomid hydrozoans only incomplete accounts exist based on the presence of either the polyp or the medusa stage. In the Atlantic and Arctic Oceans, our results linking *L. tenuis* polyps to two different hydromedusae call into question the validity of all records of *M. polydiademata* based solely on polyp morphology (*Ramil & Vervoort, 1992*; *Lundsteen, Hauksson & Gunnarson, 2020*), and cast further doubts on the previously suggested synonymy between *M. polydiademata* and the nominal species *Cuspidella grandis Hincks, 1868* (*Schuchert, 2022*). Our solution to the conundrum posed by *H. ocellata*, *L. tenuis*, and *M. polydiademata* serves as a confirmation that combining DNA barcoding, morphology and ecological information is an effective approach to link inconspicuous stages of marine invertebrates with hitherto unknown life cycles, especially in often-overlooked taxa. Furthermore, disentangling the relationships between these three taxa lays the ground for more robust analyses aimed at resolving the taxonomy and systematics of the enigmatic families Mitrocomidae and Campanulinidae.

## ACKNOWLEDGEMENTS

The authors express their thanks to the crew of RV 'Hans Brattström', Peter Schuchert, Bernard Picton, Marta Gil, the student diving club at UiB, and project MAREANO for help collecting the samples used in this study. Thanks are also due to Louise Lindblom for her help in the DNA laboratory and to Fredrik Broms for granting permission to use his images of *Halopsis ocellata*.

### Funding

The present study was funded by the Norwegian Biodiversity Information Centre through the Norwegian Taxonomy Initiative projects 70184240/NORHYDRO (LM) and 70184233/HYPNO (AH, LM), in collaboration with the Norwegian Barcode of Life (NorBOL). The funders had no role in study design, data collection, data analysis, data interpretation, or writing of the manuscript. The funders had no role in study design, data collection and analysis, decision to publish, or preparation of the manuscript.

### Grant Disclosures

The following grant information was disclosed by the authors:
Norwegian Biodiversity Information Centre through the Norwegian Taxonomy Initiative:

70184240/NORHYDRO (LM) and 70184233/HYPNO (AH, LM).
Norwegian Barcode of Life (NorBOL).

## Competing Interests

The authors declare that they have no competing interests.

## Author Contributions

- Lara M. Beckmann conceived and designed the experiments, performed the experiments, analyzed the data, prepared figures and/or tables, authored or reviewed drafts of the article, and approved the final draft.
- Joan J. Soto-Angel conceived and designed the experiments, prepared figures and/or tables, authored or reviewed drafts of the article, and approved the final draft.
- Aino Hosia conceived and designed the experiments, authored or reviewed drafts of the article, and approved the final draft.
- Luis Martell conceived and designed the experiments, authored or reviewed drafts of the article, and approved the final draft.

## Data Availability

The PCA scree plot, the Biplot, and measurement data of morphological characters, and sequences for 16S and COI are available in the Supplemental Files.

The 16S sequences are available at GenBank: OP951085–OP951093 and OQ031431–OQ031450.

The COI sequences are available at GenBank: OP945752–OP945756 and OQ031451–OQ031470.

The outgroup sequences are available at GenBank: OQ075779–OQ075782; OQ061474 and OQ061471.

Some sequences are available at BOLD: NOHYD168-21; HYPNO121-16; HYPNO144-16; HYPNO241-17; HYPNO327-18, GBCI9686-19; NOHYD024-20; NOHYD022-20; NOHYD127-21; NBCNI182-20; NOHYD125-21; NOHYD023-20; HYPNO256-17; HYPNO001-15; HYPNO296-18; HYPNO006-15; GBCI9688-19; HYPNO332-18; HYPNO180-16; HYPNO292-18; HYPNO179-16; HYPNO304-18; HYPNO339-18; HYPNO136-16; HYPNO150-16; HYPNO156-16; NOHYD020-20.

https://www.boldsystems.org/index.php/Public_RecordView?processid=NOHYD168-21.

## Supplemental Information

Supplemental information for this article can be found online at http://dx.doi.org/10.7717/peerj.15118#supplemental-information.

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
