# Peer review of "Odd family reunion: DNA barcoding reveals unexpected relationship between three hydrozoan species"

_PeerJ, doi:10.7717/peerj.15118_

## Round 0.1 · original submission · Minor Revisions

It is a high quality work, easy to read and well supported by data. The four reviewers agree and have added a few suggested changes that help make the work even better. Please take it into consideration when preparing the final version.

Reviewer 1 ·

Basic reporting

The article is of high quality with novel results and a great contribution to the study of hydrozoans. It is a clear example of integrative taxonomy and undoubtedly an example to follow for future work.

In the PDF file I placed my comments, suggestions and corrections.

Experimental design

The experimental design is state-of-the-art and considers several essential elements in the area of taxonomy, but I believe that temperature data should also be included.

Validity of the findings

All underlying data have been provided and they are robust, statistically sound. The conclusions are well stated.

Additional comments

It is important to properly cite the information contained in the introduction to avoid misinterpretation of plagiarism.

The relevance of histological and histochemical studies in taxonomy, as well as the thickness of the perisarc of the hydrotheca, can be integrated in the discussion, since from the images they seem to have differences.

Annotated reviews are not available for download in order to protect the identity of reviewers who chose to remain anonymous.

Reviewer 2 ·

Basic reporting

This is a clearly described work clarifying the systematics of several species of leptothecate hydrozoans. Specifically, the work shows beyond any reasonable doubt that hydroids that fit the morphology of Lafoeina tenuis (in family Campanulinidae), described by Sars in 1874 include at least two different species, Halopsis ocellata Agassiz, 1865 (has taxonomic priority) and Mitrocomella polydiademata
49 (Romanes, 1876; does not have priority). Both of the latter species were described based on medusa stages and are classified in the family Mitrocomidae. The work is written with clarity, and the scholarship is excellent. Similarly, the Figures illustrate the findings clearly, and the genetic data have been registered with GenBank and will presumably be released upon publication of this work.

I have just a few minor suggestions below to improve the work. The most substantive suggestion is that it would be helpful to present more detail presented on Sars' original description of Lafoeina tenuis.

Experimental design

The paper improves understanding of problems that have persisted for many decades, and improves understanding of hydrozoan systematics, squarely within the Aims and Scope of PeerJ. These problems are clearly stated and methods and analyses laid out to address them are sensical.

The morphological documentation is sound, including the statistical analyses of basic features such as polyp and nematocyst dimensions. There are a sufficient number of observations. Similarly the genetic work, focused on mitochondrial COI and 16S, and analysed in a maximum-likelihood framework for phylogenetic inference are sound and compelling, as are the species delimitation analyses (conducted in two ways).

In the descriptions of non-target taxa included in the analyses, they are labeled "As outgroups" at Line 162. However, technically not all of these taxa are outgroups (specifically, Cosmetira pilosella). Perhaps this could be re-worded as "putative and potential outgroups". This would require the issue to be brought up again briefly later in the MS when discussing the topology.

Validity of the findings

During the results, at lines 237-238, the genbank sequence JN714673 is described as "likely represents
238 a distinct independent clade", but the authors really seem to be implying (and stated elsewhere) that they are of the opinion that this sequence likely represents a distinct species.

Just after that, lines 239-240, the text states "All clades identified in the phylogenetic analyses were
recovered as putative species by the molecular species delimitation methods". However that is not quite accurate. There are clades in the COI tree (for instance within M. polydiademata) that are recovered as putative species. It might best to turn that around and state that all putative species are recovered as well supported clades, or something similar.

At line 272, the length of mastigophores is described as the most promising diagnostic character to differentiate the two main taxa being considered. Perhaps morphological character is what is meant. The genetic data and depth seem even easier characters for differentiating the two lineages.

An open question is what species was actually described by Sars in 1874. Was it Halopsis ocellata Agassiz, 1865 , Mitrocomella polydiademata (Romanes, 1876; does not have priority), or some other species? However, while I may have misunderstood the Latin, it seems likely based on depth (material from 60-300 fathoms =~110-550 meters) and locality (close by where this work was conducted) that the original description refers to Halopsis ocellata Agassiz, 1865, by priority. Thus, its polyp type is both L. tenuis-type in terms of morphology AND L. tenuis as originally described. The other lineage, Mitrocomella polydiademata (Romanes, 1876) possesses a L. tenuis-type polyp that presumably was never described. I do not think the taxonomic implications meaningfully change any of the authors conclusions. But any other potential species with L. tenuis-type hydroids are likely NOT L. tenuis and that could be more clearly stated.

Additional comments

Quick apologies to the authors and editor for being slow to accomplish my review. Things just getting away from me at the moment and it has been difficult to find the necessary time, which is also a shame because this work is great.

Finally, these problems with life cycles and tying polyps to medusa stages are even more general, applicable to Medusozoa (even conflicting taxonomies for polyps/medusae, like coronates and even Stauromedusae). I do not think the authors need to included literature review of all those cases but a reference to the larger group sharing this challenge might be nice.

In line 60, abstract "between" should be "among".

At line 77 in the introduction, the authors state that documenting hydrozoan life cycles is "further" complicated by the different life stages being primarily worked on by different researchers. But much the entire paragraph is about that and the word seems out of place. Perhaps "at least in part complicated by" might read better.

In the descriptions of the polyp stages of the two species, might it be worth adding something along the lines of "Gives rise to medusae conforming to the species X"?

Consider re-wording the Depth range descriptions in a more straight manner. e.g., Waters below 180m in region X.

Reviewer 3 ·

Basic reporting

The article meets all the criteria listed.

Experimental design

The article meets all the criteria listed.

Validity of the findings

The article meets all the criteria listed.

Additional comments

Lines 74-76: Please quote.
Lines 77-78: Please quote.
Lines 83-85: Please provide examples.
Line 328: Add space between 50 and m.
Standardize the reference format both in the in-text citations and in the reference section following PeerJ's instructions for authors.

Reviewer 4 ·

Basic reporting

The authors approach the study of three complex species, hard to identify, hard to find, tiny, and for which incomplete life cycles are known. Using molecular and morphological tools, they uncover helpful information and clarify significant taxonomic confusion about these species. The authors find that H. ocellata and M. polydiademata form two well-supported clades and that polyps of the species identified as Laofoeina tenuis fall on both clades. The results are surprising, but they highlight the major problem with hydrozoans traditionally having a split taxonomy, one for the polyp stage and one for the medusa.

Experimental design

Overall, the paper is well written, the topic is interesting, findings are relevant, and the data are solid. I thoroughly enjoyed reading it and learning how the pieces of this taxonomic puzzle fall together.
The pictures are wonderful. The paper is ready for publication.
A small suggestion:

Line 221-224- Please cite the papers/books used to identify the species.
In methods or results, I would add a sentence that describes the datasets used for each phylogenetic analysis (number of individual sequences / dataset, number of bases, etc.)

Validity of the findings

The findings are well supported by the data. The Discussion is relevant. The taxonomic picture the authors discuss is fairly complex, but they did so in a very clear way.

---

## Round 0.2 · accepted · Accept

I have read your replies to all the reviewers' comments and I'm happy with the current version of your manuscript. Therefore, it is ready for publication.